# Evaluation of In Vivo Prepared Albumin-Drug Conjugate Using Immunoprecipitation Linked LC-MS Assay and Its Application to Mouse Pharmacokinetic Study

**DOI:** 10.3390/molecules28073223

**Published:** 2023-04-04

**Authors:** Jeong-Hyeon Lim, Minjae Park, Yuri Park, Seo-Jin Park, Jiyu Lee, Sangsoo Hwang, Jeongmin Lee, Yujin Lee, Eunjeong Jo, Young G. Shin

**Affiliations:** College of Pharmacy, Chungnam National University, Daejeon 34134, Republic of Korea; jeonghyeon.lim.cnu@gmail.com (J.-H.L.);

**Keywords:** albumin-drug conjugate, glucuronidase responsive linker, quantification, LC-qTOF/MS, pharmacokinetics, bioanalytical method, protein-drug conjugates

## Abstract

There have been many attempts in pharmaceutical industries and academia to improve the pharmacokinetic characteristics of anti-tumor small-molecule drugs by conjugating them with large molecules, such as monoclonal antibodies, called ADCs. In this context, albumin, one of the most abundant proteins in the blood, has also been proposed as a large molecule to be conjugated with anti-cancer small-molecule drugs. The half-life of albumin is 3 weeks in humans, and its distribution to tumors is higher than in normal tissues. However, few studies have been conducted for the in vivo prepared albumin-drug conjugates, possibly due to the lack of robust bioanalytical methods, which are critical for evaluating the ADME/PK properties of in vivo prepared albumin-drug conjugates. In this study, we developed a bioanalytical method of the albumin-conjugated MAC glucuronide phenol linked SN-38 ((2S,3S,4S,5R,6S)-6-(4-(((((((S)-4,11-diethyl-4-hydroxy-3,14-dioxo-3,4,12,14-tetrahydro-1H-pyrano [3′,4′:6,7] indolizino [1,2-b] quinolin-9-yl)oxy)methyl)(2 (methylsulfonyl)ethyl)carbamoyl)oxy)methyl)-2-(2-(3-(2,5-dioxo-2,5-dihydro-1H-pyrrol-1-yl)-N-methylpropanamido)acetamido)phenoxy)-3,4,5-trihydroxytetra-hydro-2H-pyran-2-carboxylic acid) as a proof-of-concept. This method is based on immunoprecipitation using magnetic beads and the quantification of albumin-conjugated drug concentration using LC-qTOF/MS in mouse plasma. Finally, the developed method was applied to the in vivo intravenous (IV) mouse pharmacokinetic study of MAC glucuronide phenol-linked SN-38.

## 1. Introduction

Albumin is one of the most abundant proteins in mammalian blood, and its concentration in blood is known to be 35 to 50 mg/mL [1]. Human serum albumin consists of 585 amino acids and can be divided into three domains [2]. The essential physiological roles of albumin are to maintain the osmotic pressure of blood and bind to various biological substances such as fatty acids and hormones or exogenous substances such as drugs [3]. Most albumin is synthesized by the liver. However, its degradation mechanism is not fully understood. In general, the half-life of human serum albumin is 3 weeks, and its preferential distribution to tumor tissue has also been reported in various articles [4,5]. One suggested tumour distribution mechanism is receptor-mediated endocytosis of albumin by glycoprotein receptor (GP60 receptor or albondin) [5]. Once albumin binds to the GP60 receptor at the epithelial cellular membrane, caveolae-1 (Cav-1) mediated endocytosis vesicle formation can occur [6,7]. In addition to GP60-mediated tumor distribution, there are reports that the secreted protein acidic and rich in cysteine (SPARC) may be attributed to the tumor distribution and uptake of albumin [8,9]. Because of the aforementioned favorable pharmacokinetic characteristics (long half-life and tumor distribution) of albumin as a drug carrier, many attempts have been made to improve that of anti-cancer small molecule drugs by albumin conjugation [4,5,10,11,12,13,14]. Also, albumin binding can be used to enhance the solubility of the drug candidate. For example, Nab-paclitaxel (Abraxane), an approved albumin-bound nanoparticle drug, highly improved the solubility of paclitaxel and demonstrated its pharmacological and toxicological superiority to paclitaxel itself [15].

Not only non-covalent based albumin-bound nanoparticles, such as Nab-paclitaxel but also electrophilic functional group-based covalent conjugation strategies to endogenous albumin were devised by various pharmaceutical industries and academia. There are two minimum requirements to make this strategy come true from an anti-cancer drug development strategy perspective. First, the drug candidate must have a highly electrophilic functional group that can be conjugated with albumin’s free thiol group at its cysteine 34. Typically, in the biological environment, thiol groups can form a disulfide bond(non-reactive). However, the free thiol group of albumin Cys34 does not form a disulfide bond with another albumin and remains reactive to the exogenous electrophilic moieties such as the maleimide group [4]. Second, a tumor-selectively cleavable linker that connects the albumin-binding moiety of the drug candidate and payload should be designed to efficiently release the cytotoxic payload in the tumor.

The maleimide group is known to form a covalent bond with the reactive free thiol group as a Michael addition reaction [16,17,18] (Figure 1).

Once in vivo albumin-drug conjugate is formed, tumor specific cytotoxic payload release is required. This can be achieved by the linker, which can be cleaved by the tumor specifically. A couple of enzymes are known to be overexpressed in the tumor, such as Cathepsin B or β-glucuronidase [19,20]. For example, In several cases of ADCs, antibodies are conjugated to the payload by the Cathepsin B cleavable linker such as MC-VC-PABC linker or Glycine-Glycine-Phenlyalanine-Glycine (GGFG) tetra peptide linker [21] and the ADCs containing these linkers are already approved by the US FDA. Although there are no approved drugs that use β-glucuronidase cleavable linker yet, its selective cleavage in cancer is reported in various articles [11,22,23,24]. In addition to these enzyme-dependent linkers, acid-sensitive linkers also can be used because of the relatively low pH in the tumor microenvironment. In the case of ADC, sacituzumab govitecan (Trodelvy) has been approved using this type of CL2A linker [25]. These various kinds of ADC linkers also can be applied to albumin drug conjugate.

In this study, MAC (Methylene Alkoxy Carbamate) glucuronide linked SN-38 was selected as a “tool compound” of albumin protein-drug conjugate (Figure 2). The compound was selected based on several factors, including (1) commercial availability, (2)reactive moiety-containing linker to albumin (in this case, a maleimide group), and (3) linker stability [26]. Furthermore, (4) payload with anti-tumor activity (in this case, SN-38) etc., we have tested several compounds commercially available and found that MAC-glucuronide linked SN-38 was the best for our research. Before the pharmacokinetic studies, a reliable quantification method for the albumin drug conjugate was required. However, few bioanalytical method developments have been explored for pharmacokinetic studies because of its complex structure. Therefore, this study developed a novel bioanalytical quantification method for the “in vivo” prepared albumin-drug conjugate in mouse plasma using magnetic bead-based immunoprecipitation sample preparation and LC-qTOF/MS methods to explore the in vitro/in vivo properties of albumin-drug conjugates. Furthermore, because the developed sample preparation method can isolate the albumin from the plasma, it has a novelty in distinguishing between “albumin-conjugated” drugs and not.

The developed method was successfully applied to the in vitro mouse plasma stability study and in vivo mouse pharmacokinetic study of the albumin-drug conjugate at two different dose levels. In addition, in vivo pharmacokinetic study of SN-38 itself was also conducted to determine the pharmacokinetic improvement of the in vivo prepared albumin-drug conjugate.

## 2. Results and Discussion

### 2.1. Bioanalytical Method Development and Qualification

Immuno-precipitation method by albumin capturing magnetic bead was used to quantify albumin conjugated MAC glucuronide phenol linked SN-38. The SN-38 cleaved from the albumin-conjugated drug after incubation with β-glucuronidase was separated and quantified by the LC-MS/MS. The process of immunoprecipitation of this analytical method is shown in Figure 3b. The duplicate calibration curve standard samples were freshly prepared at each run of the assay. The calibration curve consisted of 8 points of concentration and a 50 to 20,000 ng/mL range. The quadratic regression weighted by 1/concentration^2^ was applied, and the acceptance criteria of correlation coefficient value (r) were ≥0.99. The representative calibration curve is shown in Figure 4. The analyte response at the Lower Limit of Quantification (LLOQ) was higher than five times the analyte response in the blank sample, which is sufficient for the quantification. Representative chromatograms of LLOQ and blank samples are shown in Figure 5.

Three different concentrations of QC samples (400, 2000, and 8000 ng/mL) were used for the assay performance determination. The acceptance criteria with ±25% CV were applied to assess accuracy and precision, while the acceptance criteria with ±30% CV were applied to the LLOQ [27]. The accuracy and precision results of the intra-day and inter-day assays are shown in Table 1. All bioanalytical work was done in a non-GLP environment, followed by the internal acceptance criteria mentioned above.

Various conditions of preliminary stability studies were conducted to determine whether the analyte was stable during the routine sample collection and preparation process. In addition, two hours of short-term stability study, 2 weeks of long-term stability at −80 °C and 3 cycles of freeze-thaw stability study between −80 °C and room temperature were also conducted. Each stability test condition and time interval were set based on our laboratory’s practical sample preparation time. The result of the preliminary stability study is shown in Table 2.

### 2.2. In Vitro Plasma Stability in Mouse and Human Plasma

To determine whether SN-38 is stable while conjugating with plasma albumin, in vitro stability study in mouse and human plasma at 37 °C was conducted for 7 days. Since the in vivo albumin-drug conjugate would be made after spiking the linker-payload in blank plasma, the first time point was set at 1 h after spiking in order to have sufficient time for bio-conjugation of the linker with albumin in blank plasma. In mouse plasma, 80% of MAC glucuronide phenol linked SN-38 was still conjugated with albumin after 7 days of incubation compared to the initial timepoint sample. In the case of human plasma, no significant concentration change was observed after 7 days of incubation. The results indicated that albumin-conjugated MAC glucuronide phenol linked SN-38 is quite stable in both species plasma. The results are shown in Figure 6.

There might be a possibility of the retro-Michael addition reaction, which can cleave the albumin-drug conjugation. However, our results indicated that once albumin-drug conjugate was formed, the undesired de-conjugation by the retro-Michael addition would not happen. It also implied that the albumin-drug conjugate would likely be stable during systemic circulation if it were formed in vivo [28]. This result also indicated that the β-glucuronidase dependent cleavable linker used in this study was stable for 7 days under incubation.

### 2.3. Application for In Vivo Mouse Pharmacokinetic Study

One mg/kg and 3 mg/kg of the MAC glucuronide phenol linked SN-38 were administered to ICR mice by single intravenous (IV) for mouse pharmacokinetic study (as an equivalent dose of SN-38, 0.36 and 1.08 mg/kg respectively, *n* ≥ 3). Plasma samples obtained from this study were analyzed using the developed bioanalytical method. In addition, one mg/kg of pharmacokinetic study of SN-38 was also conducted, and the samples were analyzed using the protein precipitation method. As a result, the pharmacokinetic profiles of the albumin-conjugated SN-38 and the SN-38 itself are shown in Figure 7, and its pharmacokinetic parameters are also shown in Table 3.

Half-life (T_1/2_); maximum concentration (C_max_); area under the plasma concentration vs. time curve from 0 to last time point (AUC_last_); area under the curve from time zero to infinity (AUC_INF_); systemic clearance (CL); the volume of distribution at steady state (V_ss_).

The half-life and volume of distribution between IV 1 mg/kg and 3 mg/kg of MAC glucuronide phenol linked SN-38 were comparable. The dose proportionality evaluation of the C_max_ and exposure were conducted using the power regression model. The exploratory dose proportionality was confirmed in this dose range, and the result is shown in Figure 8. Although SN-38 showed relatively high in vivo clearance and was fully eliminated in one day, the albumin-conjugated SN-38 showed very low clearance and a much smaller volume of distribution. This PK result implies that MAC glucuronide phenol linked SN-38 would be able to conjugate albumin efficiently “in vivo” and follows the PK behaviours of albumin with a long half-life. The similarity between the known half-life (around 35 h) of mouse albumin and the results of this study also supports our hypothesis [29].

## 3. Materials and Methods

### 3.1. Materials

MAC glucuronide phenol linked SN-38, and SN-38 was purchased from MedChem Express (Monmouth Junction, NJ, USA). Verapamil was used for an internal standard (ISTD) for albumin conjugated drug and free payload quantification assay, and β-glucuronidase was purchased from Sigma-Aldrich (St Louis, MO, USA). Albumin magnetic bead was purchased from Millipore Korea (Daejeon, Republic of Korea). Blank ICR mouse and IgG-depleted human plasma were purchased from BioMedex Korea (Seoul, Republic of Korea). Other reagents were commercially purchased for analytical purposes or reagent grading and were used without further purification.

### 3.2. Preparation of Stocks, Standard (STD) and Quality Control (QC) Samples

A stock solution of MAC glucuronide phenol linked SN-38 was prepared by dissolving it in dimethyl sulfoxide (DMSO) at 1 mg/mL concentration. Then the 0.1 mg/mL sub-stock solution was prepared by dissolving in DMSO and then serially diluted with DMSO to prepare the calibration standard working solutions. Next, an aliquot of 4 µL of each calibration standard (STD) and quality control (QC) solution was spiked to 20 µL of mouse blank plasma. All the STD and QC samples were treated as same as the study sample preparation procedures. The calibration curves were constructed in the y-weighted quadratic regression (1/x^2^) with a correlation coefficient (r) value ≥ 0.99 ranging from 50 to 20,000 ng/mL. Assay performance was assessed by the accuracy and precision of QC samples with three concentrations (400, 2000, and 8000 ng/mL).

### 3.3. Sample Preparation

Each 24 µL of STD and QC plasma sample was mixed with 350 µL of 0.1% tween 20 in PBS and 20 µL of albumin magnetic bead. In addition, 20 µL of all study samples were also mixed with 4 µL of DMSO as make-up and 350 µL of 0.1% tween 20 in PBS and 20 µL albumin magnetic bead. After vortexing, mixture samples were incubated gently for about 2 h at room temperature, and then samples were fixed on a magnetic rack and washed twice, first washed with 200 µL of 0.1% tween 20 in PBS and then the second with 200 µL of PBS.

To quantify the total amount of SN-38 conjugated to albumin, the PBS-washed bead samples were treated with 30 µL of 0.1 mg/mL β-glucuronidase in PBS and incubated for 12 h at 37 °C. Next, the mixture was quenched with 4 µL of 2N HCl to deactivate the enzyme activity. Thirty microliters of ACN containing 20 ng/mL verapamil as an internal standard were added to extract SN-38. Finally, the samples were centrifuged at 4500× *g* for 5 min, and then 50 µL of supernatant was transferred into an LC vial for LC-qTOF/MS analysis. The enzymatic release mechanism of SN-38 from the MAC glucuronide phenol linker when incubated with β-glucuronidase is shown in Figure 9 [30,31,32].

### 3.4. Method Qualification

A fit-for-purpose method qualification was carried out using LC-qTOF/MS. The qualification run contained a duplicate of eight concentrations of standards and quality control samples at three different concentration levels. The in-house acceptance criteria for STDs and QCs in the qualification run were within ±25% of the nominal values for the accuracy and precision values [27]. A quadratic regression (weighted 1/concentration^2^), with an equation of y = ax^2^ + bx + c, was used to fit the calibration curves. In addition, two blank plasma samples were also run, one with blank plasma with ISTD and the other with double blank plasma (no ISTD and no analyte). The intra- and inter-day runs for the accuracy and precision assays were calculated at each QC concentration.

Various conditions of stability tests were conducted in mouse plasma samples. Three different concentrations of QC samples were tested for short-term, long-term, and freeze-thaw stability. At room temperature, the short-term stability was determined for 2 h. Long-term stability was determined by analyzing QC samples frozen at −80 °C for 2 weeks. Three freeze and thaw cycles at −80 °C were conducted to know whether the QC samples were stable. Each stability condition was set based on the times and cycles of routine PK sample preparation procedures. The acceptance criteria for the accuracy and precision of the preliminary stability test were also set within ±25% based on the in-house criteria.

### 3.5. In Vitro Mouse and Human Plasma Stability

For the in vitro linker stability test, 10 µL of 1.0 mg/mL of MAC glucuronide SN-38 in DMSO was spiked into 990 µL of a mouse or human blank plasma to make an incubation samples. After incubation at 37 °C for 1 and 4 h; 1, 2, 4 and 7 days, the incubated samples were stored at −80 °C until LC-MS/MS analysis.

### 3.6. Application for Preclinical In Vivo Mouse Pharmacokinetic Study

The preclinical pharmacokinetic studies were conducted in ICR mouse. MAC glucuronide phenol linked SN-38 was directly administered to the mouse via single intravenous bolus injection at 1 and 3 mg/kg (as an equivalent dose of SN-38, 0.36 and 1.08 mg/kg, respectively). Blood samples were collected through an infraorbital vein at 5 min; 1, 4, and 7 h; 1, 2-, 4-, 7- and 10-days post-dose using a heparinized capillary tube. The blood samples were centrifuged, and the supernatant plasma was stored at −80 °C until LC-MS/MS analysis. Animal experiments followed the animal care protocol (Protocol code 202103A-CNU-054) approved by the Chungnam National University. All procedures related to animal experiments were also performed per the guidelines established by the Association for Assessment and Accreditation of Laboratory Animal Care International (AAALAC International). PK parameters were calculated by non-compartmental analysis (NCA) using WinNonlin^®^ version 8.3.4 (Pharsight Corporation, Mountain View, CA, USA). Dose proportionality was estimated using the power regression model. This model defines the relationship between the dose and pharmacokinetic parameters,, especially for C_max_, AUC_last_, and AUC_INF,_ as the following equation [33,34].
$$log\left(PK\;parameter\right)=\alpha +\beta log\left(dose\right)$$

In this equation, if the range of slope of the regression line (*β*) with a 90% confidence interval is included in the pre-specified range, the dose proportionality can be claimed. In this study, the pre-specified criterion suggested by Hummel et al. was applied [35].

### 3.7. LC-qTOF/MS Conditions

Two Shimadzu LC-20AD pumps, a Shimadzu CBM-20A HPLC pump controller (Shimadzu Corporation, Columbia, MD, USA), a CTC HTC PAL autosampler (LEAP Technologies, Carrboro, NC, USA), and a quadrupole time-of-flight TripleTOF™ 5600 mass spectrometer (Sciex, Foster City, CA, USA) equipped with a Duospray™ ion source (Sciex, Foster City, CA, USA) were used for LC-qTOF/MS analysis.

The binary flow pumping mode of two Shimadzu LC-20AD pumps was used. Distilled and deionized water containing 0.1% formic acid was used as a mobile phase A and acetonitrile containing 0.1% formic acid was used as a mobile phase B. As a result, the initial mobile phase B percent of gradient elution increased from 10% mobile phase B to 95% mobile phase B during the run time. The LC gradient for this analysis is summarized in Table 4. The injection volume was 10 µL, and the LC flow rate was 0.4 mL/min. The SN-38, which would be released from the albumin-conjugated drug, was separated through a Phenomenex Kinetex XB-C18 column (2.1 × 50 mm, 2.6 µm). The column temperature was maintained at 55 °C.

The TOF-MS and product ion scans mass spectra were recorded in the positive ion mode. The scan range was set at *m*/*z* 100~900 in the TOF-MS scan and at *m*/*z* 100~700 in the product ion scan. For the quantification, [M + H]+ parent ions of SN-38 and verapamil were selected at *m*/*z* 393.1 and *m*/*z* 455.3, respectively, and their product ions at *m*/*z* 349.2 and *m*/*z* 165.1 were used for the quantitative analysis, respectively. The ion source temperature was 500 °C, and the ion spray voltage was 5500 V. For the SN-38 and verapamil, the declustering potentials were 150 V and 125 V, respectively, and the collision energies were 40 V and 30 V, respectively. Sufficient system equilibrium was conducted before the analytical run was operated. In addition, the mass spectrometer was calibrated on the TOF-MS in the positive mode using a calibration solution standard for TripleTOF™ 5600 with an error ppm less than 1 ppm. Analyst^®^ TF 1.6 software (Sciex, Foster City, CA, USA) was used for the data acquisition and analysis.

## 4. Conclusions

The albumin-drug conjugate has been studied in academia and the pharmaceutical industries since its concept was proposed. Most the researchers have explored it either from efficacy or reaction mechanism perspectives. However, few studies have been conducted for the albumin-drug conjugate regarding the bioanalytical methods and the pharmacokinetic analysis perspectives, particularly for the in vivo prepared-albumin drug conjugate. This study successfully developed a novel bioanalytical method using a magnetic bead-based immuno-precipitation followed by LC-qTOF/MS analysis to investigate an “in vivo” prepared albumin-drug conjugate. The accuracy and precision of the developed method met our in-house discovery bioanalytical acceptance criteria. They were satisfactory for various in vitro stability assays and in vivo mouse PK studies. The dose-proportional pharmacokinetic properties of albumin-MAC glucuronide phenol linked SN-38 were observed from the single IV bolus at 1 and 3 mg/kg. Since this research was still at the early stage of understanding the “in vivo prepared” albumin conjugated drug, no higher doses or multiple doses with various dose intervals were investigated yet in this study. Nevertheless, considering the high concentration of in vivo albumin (35~50 mg/mL) in blood, albumin is a good conjugation target for “in vivo prepared” albumin-conjugated drugs. Therefore, it would be hardly saturated at conventional therapeutic doses unless very high toxic doses were administered. In addition, the half-life of human serum albumin is also known to be 3 weeks, much longer than that of murine serum albumin. If so, this approach would be more attractive to the pharmaceutical industry from the clinical pharmacokinetic perspective. Further studies using xenograft mouse models with various multiple dosing regimens would be warranted to understand its pharmacokinetic and pharmacodynamic correlation and the preclinical/clinical development potentials of the “in vivo prepared” albumin drug conjugate approach.

## Figures and Tables

**Figure 1 molecules-28-03223-f001:**
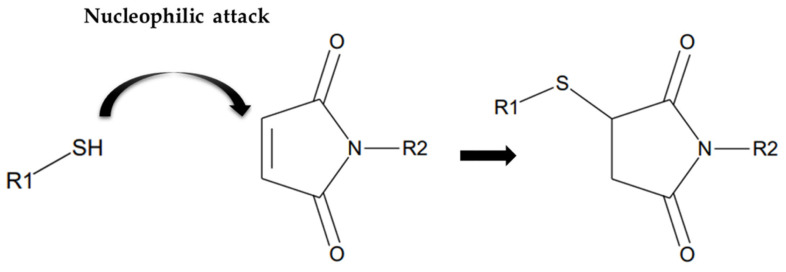
Scheme of Michael addition reaction between free sulfhydryl group and maleimide group.

**Figure 2 molecules-28-03223-f002:**
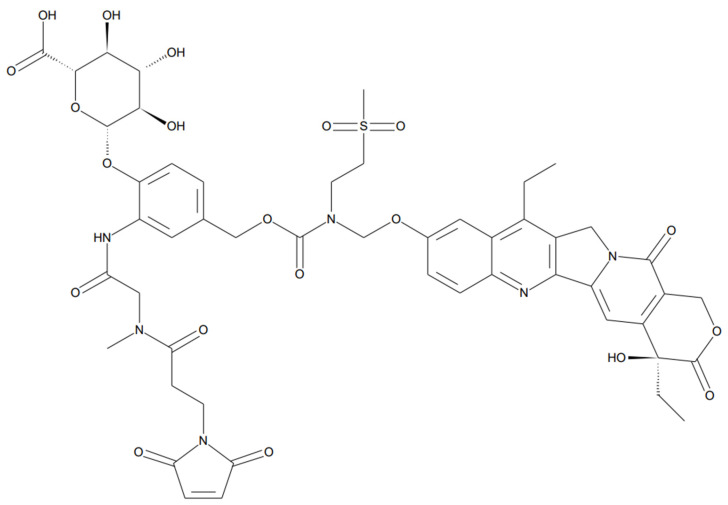
Structure of MAC glucuronide phenol linked SN-38 ((2S,3S,4S,5R,6S)-6-(4-(((((((S)-4,11-diethyl-4-hydroxy-3,14-dioxo-3,4,12,14-tetrahydro-1H-pyrano [3′,4′:6,7]indolizino [1,2-b]quinolin-9-yl)oxy)methyl)(2-(methylsulfonyl)ethyl)carbamoyl)oxy)methyl)-2-(2-(3-(2,5-dioxo-2,5-dihydro-1H-pyrrol-1-yl)-N-methylpropanamido)acetamido)phenoxy)-3,4,5-trihydroxytetra-hydro-2H-pyran-2-carboxylic acid).

**Figure 3 molecules-28-03223-f003:**
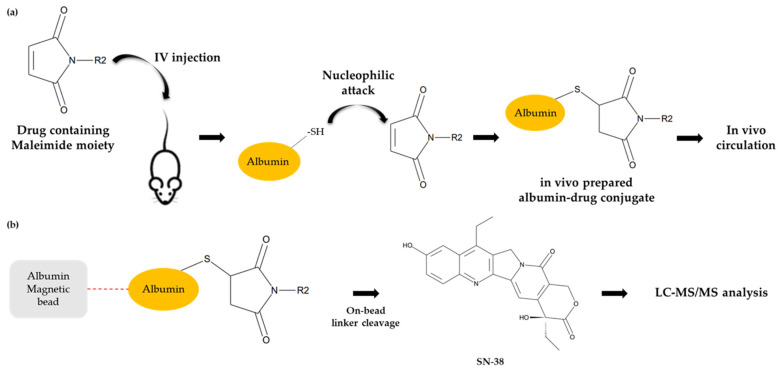
Conceptual scheme of albumin drug conjugate (**a**) and the brief process of the developed analytical method (**b**).

**Figure 4 molecules-28-03223-f004:**
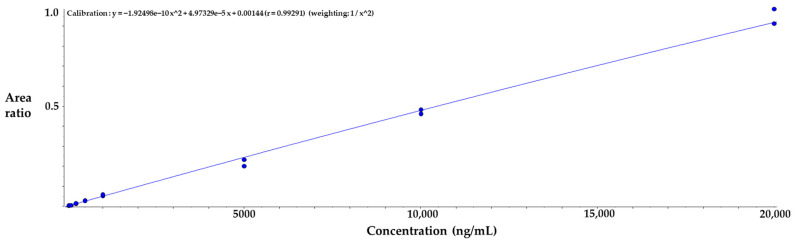
Representative calibration curve of SN-38 released from MAC glucuronide phenol linked SN-38.

**Figure 5 molecules-28-03223-f005:**
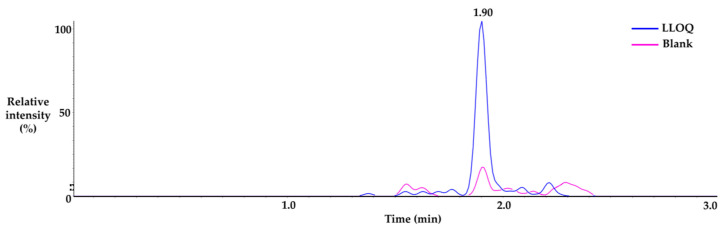
Representative chromatogram of LLOQ and blank sample.

**Figure 6 molecules-28-03223-f006:**
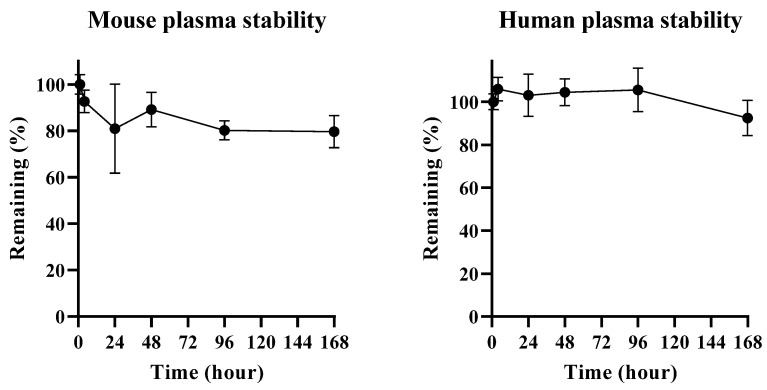
Plasma stability test result of albumin conjugated MAC glucuronide phenol linked SN-38 in mouse and human plasma.

**Figure 7 molecules-28-03223-f007:**
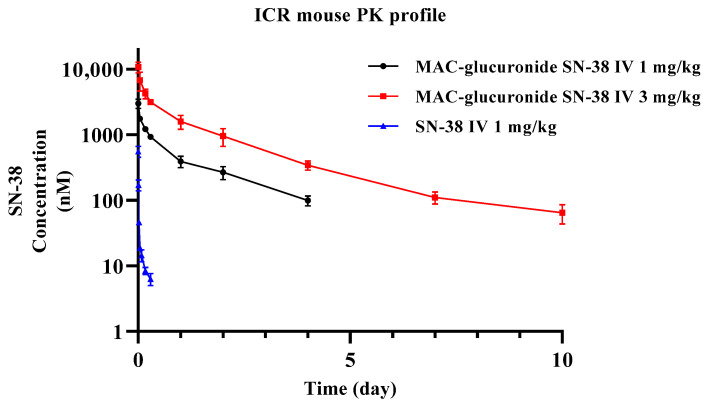
Pharmacokinetic profiles of SN-38 at 1 mg/kg and albumin conjugated SN-38 at 1 mg/kg and 3 mg/kg MAC glucuronide phenol linked SN-38 after Intravenous administration.

**Figure 8 molecules-28-03223-f008:**
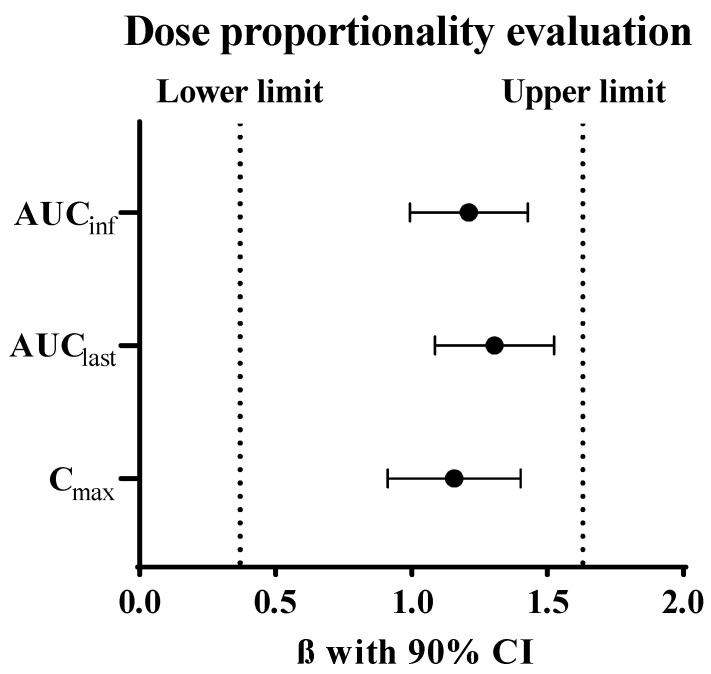
Dose proportionality evaluation between 1 and 3 mg/kg of MAC glucuronide phenol linked SN-38 using the power regression model.

**Figure 9 molecules-28-03223-f009:**
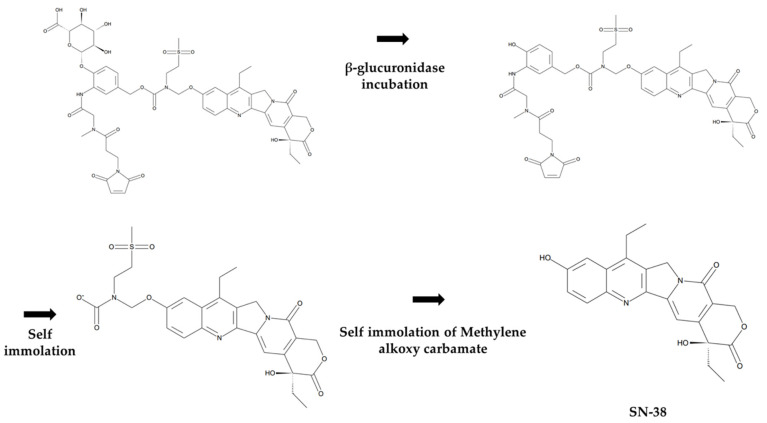
Mechanism of SN-38 release from MAC glucuronide phenol linked SN-38 after incubation with β-glucuronidase.

**Table 1 molecules-28-03223-t001:** Intra/inter-day accuracy and precision of MAC glucuronide phenol linked SN-38 in quality control samples at three levels (*n* = 3 for intra-day accuracy and precision; *n* = 9 for inter-day accuracy and precision).

Intra-Run Assay				
Run No.	Statistics	Low QC (400 ng/mL)	Medium QC (2000 ng/mL)	High QC (8000 ng/mL)
1	Mean concentration (ng/mL)	417.25	1994.94	7487.20
	Accuracy (%)	104.31	99.75	93.59
	Precision (%, CV)	15.07	21.36	3.58
2	Mean concentration (ng/mL)	447.86	2090.10	8071.21
	Accuracy (%)	111.96	104.50	100.89
	Precision (%, CV)	10.77	6.31	17.29
3	Mean concentration (ng/mL)	394.98	1867.89	7990.39
	Accuracy (%)	98.75	93.39	99.88
	Precision (%, CV)	15.02	4.73	18.32
Inter-Run Assay				
Run 1~3	Mean concentration (ng/mL)	420.03	1984.31	7849.60
	Accuracy (%)	105.01	99.22	98.12
	Precision (%, CV)	12.25	11.74	12.69

**Table 2 molecules-28-03223-t002:** The stability assessment of MAC glucuronide phenol linked SN-38 in mouse plasma (*n* = 3).

Assessment	Statistics	Low QC (400 ng/mL)	Medium QC (2000 ng/mL)	High QC (8000 ng/mL)
Short-term stability	Mean concentration	362.18	1917.73	8152.33
	Accuracy (%)	90.54	95.89	101.90
	Precision (%, CV)	12.15	17.46	6.62
Long-term stability	Mean concentration	446.90	1724.19	6159.21
	Accuracy (%)	111.73	86.21	76.99
	Precision (%, CV)	8.23	3.17	3.02
Freeze-thaw stability	Mean concentration	484.37	1969.98	8172.28
	Accuracy (%)	121.09	98.50	102.15
	Precision (%, CV)	1.84	8.64	12.78

**Table 3 molecules-28-03223-t003:** Pharmacokinetic parameters of SN-38 at 1 mg/kg and albumin conjugated SN-38 at 1 mg/kg and 3 mg/kg of MAC glucuronide phenol linked SN-38 (as an equivalent dose of SN-38, 0.36 and 1.08 mg/kg respectively) after intravenous administration.

PK Parameters						
	T_1/2_ (min)	C_max_ (ng/mL)	AUC_last_ (min × ng/mL)	AUC_INF_ (min × ng/mL)	CL (mL/min/kg)	V_ss_ (mL/kg)
MAC glucuronide phenol linked SN-38 IV 1 mg/kg	2187.81	3291.42	2,495,425.63	2,845,522.93	0.35	863.47
SD	531.83	529.00	236,162.83	209,510.79	0.03	163.66
MAC glucuronide phenol linked SN-38 IV 3 mg/kg	2986.94	11,772.15	10,577,105.23	10,884,918.68	0.28	818.41
SD	822.30	2030.98	1,777,851.60	1,782,896.03	0.04	138.86
SN-38 IV 1 mg/kg	239.19	218.71	4357.47	5225.42	191.49	36,420
SD	34.23	41.04	288.08	28.32	1.04	9470

**Table 4 molecules-28-03223-t004:** The LC gradient elution for the separation of SN-38.

Time (min)	Mobile Phase B (%)
0.0	10
0.5	10
0.95	95
1.4	95
1.5	10
3.0	10

## Data Availability

The data presented in this study are available on request from the corresponding author.

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
