# Peer review of "Evaluation of In Vivo Prepared Albumin-Drug Conjugate Using Immunoprecipitation Linked LC-MS Assay and Its Application to Mouse Pharmacokinetic Study"

_molecules, 2023, doi:10.3390/molecules28073223_

Round 1

Reviewer 1 Report

Enhancing the selectivity of anticancer drugs currently used in the clinic is of great interest in order to propose more efficient chemotherapies with fewer side effects for patients.  Recently, it is shown that β-glucuronidase-responsive albumin-binding prodrugs can be a valuable alternative to internalising drug delivery systems for the treatment of solid tumours [Renoux B., et al. 2017; Renoux B., et al., 2018]. These prodrugs are composed of a glucuronide trigger, a potent anticancer drug and a self-immolative linker bearing a hydrophilic side chain with a maleimide functional group at the end. Once in the bloodstream, such molecular assemblies bind covalently to plasmatic albumin and accumulate passively in tumours where extracellular β-glucuronidase triggers the release of the active compound. These results can be of interest for further development in this field, aiming to enhance the efficacy of targeted cancer chemotherapy. However, the present work is not a significant contribution to the field.

Introduction is not in line with the discussed content. The introductory section is bad written in terms of language. Moreover, it is out of focus and does not match the content of the remainder of the manuscript, nor is it aligned with the goals that are put forward. This introduction needs to reviews the pharmacokinetics of different water-soluble conjugates of irinotecan, its more potent metabolite SN-38, and SN-38 glucuronide, interfere with mammalian DNA topoisomerase I and cancer cell death appears to result from DNA strand breaks caused by the formation of cleavable complexes. However, it focuses on the molecular and practical aspect of the properties of human serum albumin. Moreover, the authors hypothesized that a maleimide reagent would lead to bioconjugates featuring enhanced stability (Fig 1). Indeed, maleimide chemistry stands out in the bioconjugation toolbox by virtue of its synthetic accessibility, excellent reactivity, and practicability. Nonetheless, most of these bioconjugation reactions are suboptimal as they cannot reach full conversion. Addition of an excess of reagents is commonly used as a mitigating action, though often las to over-modification and/or erosion of the chemospecificity (e.g. maleimide cross reactivity with lysines). HSA has 35 cysteine residues; 34 are paired in 17 disulfide bonds leaving only Cys34 available for site-specific chemical modification. However, DTNB (5,5’-dithio-bis(2-nitrobenzoic acid)) titrations indicate the sulfhydryl titer for most commercial plasma HSA preparations is approximately 30–40%. [H. Era, et al. 2013; S. Miyamura, et al. 2016]. The heterogeneity of HSA is central to its physiological role and presents a complication for protein modification, but that is the biological reality. Maleimide reagents was found to conjugate not only to free SH of cysteine residue but also to lysine residues in albumin [Kikuchi, S., et al., 2016;  Ishii, S., et al., 2019]. There are as many as 59 lysine residues in albumin, providing 59 amine groups as potential modification sites by maleimide reagents. So, the amine and sulfhydryl groups in albumin could be used for HSA modification.

Not all the cited references are   appropriate and adequate (see, for example, 1-3).

The chapter 2 (Results and Discussion) aims to summarize the results regarding investigation of in vivo prepared albumin-drug conjugate using immunoprecipitation linked LC-MS assay and its application to mouse pharmacokinetic study.  However, a meaningful discussion of these challenges is lacking. While the synthetic procedures and the accompanying observations in the chapters 2.2 and 2.3 are certainly interesting, they do not add sufficient weight to justify publication as such in Molecules. To figure out whether SN-38 is stable while conjugating with plasma albumin, in vitro stability study in IgG depleted human plasma. However, conjugate can be undergoes retro and exchange reactions in the presence of other thiol compounds, such as IgG at physiological pH and temperature, offering a novel strategy for controlled release.  Retro-Michael and thiol-exchange reactions in the presence of biothiols under physiological conditions have been clearly demonstrated by Kiick and co-workers [Baldwin, A.D.; Kiick, K.L., 2011].

In the conclusions it was preposed that further studies using xenograft mouse models with various multiple dosing regimens would be warranted to understand its preclinical/clinical development potentials of the “in vivo prepared” albumin drug conjugate approach. However, xenograft models are based on the implantation of human tumor cells into immunocompromised mice to avoid graft versus host reaction of the mouse against the human tumor tissue. It has been previously observed in mice that serum half-life of administered human albumin was strikingly short (2 d) [Roopenian, D.C., et l. 2015]. This is in contrast to the 21 d half-life of HSA observed in humans and hence questions the translational value using xenograft models for modeling HSA pharmacokinetic behavior. The proposed cause for this are is species differences, in which mouse FcRn (mFcRn) binds HSA poorly, and therefore is being outcompeted by the excessive concentrations of mouse serum albumin in mice, leading to its short half-life. A major limitation to preclinical pharmacokinetic evaluation and development of novel serum albumin-based therapeutics has been the lack of animal models that more faithfully reflect human albumin metabolism.

Reviewer 2 Report

Manuscripts are well written and English is well established. I suggest adding one more image, a kind of graphical abstract, but placed in the discussion. This will make it easier to understand the contents of the paper and increase readability. Furthermore, it is highly recommended to increase the resolution of the current figures. All references < 2010 year of publish, it is strongly recommended to change to the latest (>2010).

I hope authors will do that.

Reviewer 3 Report

The manuscript describes and discusses logically designed experiments and presents results that are expected to be of large interest for the scientific community. It is an interesting study with an interesting approach. The paper in the whole is well designed and results sound. Nevertheless, the manuscript needs a minor revision:

Point 1: In the introduction part should be more highlighted the main aim of the paper, and additionally, what is the novelty of carried research work.

Point 2: How do the Authors select the analytes? The rational of the choice of the selected biologically active compounds studied is missing and should be clearly discussed.

Point 3: Quality of the figures must be improved.

Round 2

Reviewer 1 Report

The authors' answers satisfied the reviewer. Per the reviewer’s comments, the introduction as well as the results/discussion sections have been revised accordingly. In my opinion manuscript could be accepted for publication in Molecules upon minor revision.

Minor point.

Authors cited the references 1 and 2 because of their descriptions about albumin’s physiological roles and its general structural information. Reference 1 said “Serum albumin is the most abundant protein in the blood plasma of all vertebrates with the concentration in human serum being 35–50 mg/mL”. Reference 2 said that “Human serum albumin (HSA) has a plasma concentration of 35–50 mg/mL”. However, in these articles there is reference to an earlier article (Peters T., 1996). It is in it said that “Serum albumin is the most abundant protein in the blood plasma of all vertebrates with the concentration in human serum being 35–50 mg/mL”. Therefore, it is necessary to replace reference 1 with more appropriate (Peters, T. All About Albumin: Biochemistry, Genetics and Medical Application; Academic Press Limited: Cambridge, MA, USA, 1996).
